# Canna Starch Improves Intestinal Barrier Function, Inhibits Allergen Uptake, and Suppresses Anaphylactic Symptoms in Ovalbumin-Induced Food Allergy in Mice

**DOI:** 10.3390/biom14020215

**Published:** 2024-02-12

**Authors:** Ayaka Koida, Mamoru Tanaka, Rina Kosaka, Shoei Okuda, Shiro Takei, Suzuno Ota, Sayaka Yokoyama, Kaho Miyake, Hiroyuki Watanabe

**Affiliations:** 1Faculty of Health and Medical Sciences, Aichi Shukutoku University, 2-9 Katahira, Nagakute 480-1197, Aichi, Japan; akoida@asu.aasa.ac.jp; 2Graduate School of Bioscience and Biotechnology, Chubu University, 1200 Matsumoto, Kasugai 487-8501, Aichi, Japan; gr22017-7137@sti.chubu.ac.jp (R.K.); gr23013-4634@sti.chubu.ac.jp (S.O.); stakei@isc.chubu.ac.jp (S.T.); gr20028-7827@sti.chubu.ac.jp (K.M.); 3College of Bioscience and Biotechnology, Chubu University, 1200 Matsumoto, Kasugai 487-8501, Aichi, Japan; 4Faculty of Health Science, Suzuka University of Medical Science, 1001-1 Kishioka, Suzuka 510-0293, Mie, Japan; suzuno-o@suzuka-u.ac.jp; 5Department of Food and Nutritional Environment, Kinjo Gakuin University, 2-1723 Omori, Moriyama-ku, Nagoya 463-8521, Aichi, Japan; yokoyama-s@kinjo-u.ac.jp; 6Faculty of Nutrition, University of Kochi, 2751-1 Ike, Kochi 781-8515, Kochi, Japan; watana@cc.u-kochi.ac.jp

**Keywords:** canna starch, type 1 allergy, intestinal barrier function, anaphylactic symptom, mouse

## Abstract

Edible canna rhizomes contain extremely high levels of resistant starch among cereals and potatoes. We previously showed that feeding canna rhizome starch to mice may increase intestinal barrier function and improve the intestinal environment. Here, we investigated the effects of canna starch intake in a murine food allergy model. Five-week-old female BALB/c mice were divided into four groups: Control and OVA groups fed on the control diet (AIN-93G) ad libitum and Canna and OVA-Canna groups fed on the canna diet (AIN-93G with 10% replaced with canna starch). The OVA and OVA-Canna groups were sensitized to ovalbumin (OVA), and the anaphylactic response was assessed by measuring body temperature. Body temperature was significantly lower in the OVA group than in the non-sensitized group, but no decrease was observed in the OVA-Canna group. Fecal weight, fecal mucin content, and goblet cells of colorectal tissue were significantly increased in the Canna and OVA-Canna groups compared with those in the Control and OVA groups. Allergen uptake into the liver was also increased in the OVA group and decreased in the OVA-Canna group to the same level as in the non-sensitized group. These results indicate that canna starch supplementation in a murine food allergy model suppresses anaphylactic symptoms by improving the intestinal environment and reducing allergen uptake by increasing intestinal barrier function.

## 1. Introduction

In recent years, the number of patients with type 1 allergic diseases, including hay fever and food allergy, has continued to increase in developed countries, including Japan [1,2]. Various symptoms are induced by food allergies, including serious ones such as anaphylactic shock involving decreased blood pressure and loss of consciousness. The current approach to managing food allergies involves strategies such as consuming an elimination diet that excludes the causative food, acquiring tolerance through oral immunotherapy, and using medications such as histamine H1 receptor antagonists and corticosteroids [1]. However, concerns have been raised that elimination diets reduce quality of life due to the physical (e.g., purchasing food, meal planning, and cooking) and emotional burdens (e.g., feeling uneasy about the feedback from others and being unable to enjoy meals with children) [3], oral immunotherapy may induce symptoms, and drugs are only a symptomatic treatment to improve allergic symptoms and do not prevent or cure the disease. Moreover, the ingestion of anti-allergy drugs or foods mainly targets mast cells, IgE, and degranulation and is intended for people who are already allergic. Meanwhile, to prevent allergies altogether, it is important for individuals not to be in an allergic state or not to develop allergic symptoms.

While several reports about the suppression of anaphylactic symptoms by food components have been published [4,5], the detailed mechanisms associated with this have not been fully elucidated. Nevertheless, this approach has numerous advantages, including fewer side effects, the synergistic effects of multiple components, and the ability to incorporate these components into the daily diet. There are high expectations for foods that can prevent the onset and severity of food allergies through their regular consumption as part of the diet.

Edible canna (*Canna edulis*) is a generic term for a large perennial plant native to Central and South America, belonging to the Cannaceae family of ginger plants, with its enlarged rhizome serving as a source of food due to its high starch content. The main sources of starch in the human diet are cereal grains (corn, wheat, and rice) and rhizomes (tapioca, sweet potato, and potato). However, it has been reported that canna starch has relatively low digestibility, at around 53%, compared with the digestibility of raw starch from potatoes and cassava, reaching over 90% [6]. In general, starch grains with a smaller diameter are more easily degraded by enzymes due to the larger surface area per unit weight, but canna starch is characterized by very large oblong granules, relatively high amylose content, and high thermal resistance to viscosity breakdown compared with corn and potato starch [7]. As a functional food, *C. edulis* has been reported to exert antioxidant [8], anti-obesity [9], anti-diabetic [10], and anti-hyperlipidemic effects [9], as well as being able to inhibit carcinogenesis [11]. We also compared canna starch with starch from five other sources (wheat, sweet potato, ginger, corn, and potato) to characterize its constituents and found it to be rich in amylose and resistant starch [12]. Furthermore, BALB/c mice fed canna starch showed a significant increase in bacteria of the *Clostridium* genus [12], which are representative bacteria that induce the differentiation of regulatory T (Treg) cells, the control cells of the immune response [13,14,15]. These mice also exhibited an increase in organic acids in the cecal contents, such as lactic acid, acetic acid, and n-butyric acid, which are metabolic products of intestinal bacteria. In the feces of these mice, we also observed elevated levels of mucins and IgA, which are involved in intestinal defense functions [12]. In the intestinal lumen, mucins and IgA play crucial roles in maintaining the integrity of the intestinal barrier. They are responsible for eliminating foreign substances such as allergens and viruses [16]. Furthermore, an increase in Treg cells is anticipated to be beneficial in conditions characterized by excessive immune responses, such as allergies, atopic dermatitis, colorectal cancer, and inflammatory bowel diseases [17]. These results suggested that canna starch is effective at preventing and improving immune disorders such as allergic diseases. Against this background, we investigated the preventive effect of canna starch on food allergies and the associated mechanisms, especially on the intestinal lumen.

## 2. Materials and Methods

### 2.1. Canna Starch Preparation

Canna starch obtained from the Watanabe laboratory at the University of Kochi was extracted by first suspending it in water and then drying it, which was comparable to the conventional method [12].

### 2.2. Animals and Diets

Female 5-week-old BALB/cCrSlc mice (15–20 g) were obtained from Japan SLC (Hamamatsu, Japan) and housed in a room at 22 ± 2 °C and 50 ± 10% relative humidity with a 12 h light–dark cycle in a specific pathogen-free facility. After acclimation for 7 days, the mice were provided with experimental diets and water *ad libitum*. On the day before the start of the experiment, all mice were weighed and divided into four groups: Control group (n = 8), Canna group (n = 10), OVA group (n = 10), and OVA-Canna group (n = 9), ensuring equal body weights among the groups. The experimental design was in accordance with the guidelines for animal experimentation and approved by the Animal Experimentation Committee of Chubu University (authorization number: 20191004).

The experimental diets shown in Table 1 were adjusted as described previously [12]. For 28 days during the experimental period, the Control group and OVA group were fed control diets (AIN-93G) ad libitum, while the Canna group and OVA-Canna group were fed canna diets (AIN-93G with 10% replaced with canna starch) ad libitum.

### 2.3. Sensitization

The mice were sensitized to OVA as described previously [18] with some modifications. In brief, the sensitized mice (OVA group and OVA-Canna group) were injected intraperitoneally with 50 μg of ovalbumin (OVA, A5503; Sigma-Aldrich, St. Louis, MO, USA) and 4 mg of aluminum hydroxide (Imject™ Alum; Thermo Fisher Scientific, Waltham, MA, USA) emulsified in 0.2 mL of phosphate-buffered saline (PBS), pH 7.0, on days 7 and 21. The non-sensitized animals (the Control group and the Canna group) received aluminum hydroxide in PBS as a vehicle control.

### 2.4. Animal Protocol

Determination of growth performance over the feeding period was evaluated using body weight on days 0 and 28. To assess intestinal barrier function, mice were divided into individual cages, and fecal samples were collected for 72 h on days 25–28, after which they were stored at −80 °C and then freeze-dried and milled. Twenty-eight days after the beginning of the experiment, all mice (10 weeks old) were orally administered 20 mg of OVA in 0.2 mL of PBS. Thirty minutes later, the rectal temperature was measured using an endorectal probe and AD-1687 Weighing Environment Logger (A&D Co., Tokyo, Japan) to evaluate allergic responses. The mice were anesthetized with isoflurane and then sacrificed to collect the liver and colon. The liver and colon were fixed in 4% paraformaldehyde phosphate buffer solution (FUJIFILM Wako Pure Chemical Industries, Osaka, Japan) at 4 °C. The fixed liver was used for immunostaining, and the fixed colon was subjected to Alcian Blue-PAS staining.

### 2.5. Analyses of Fecal-Specific IgA, Total IgA, and Mucin

Total IgA and mucin assays were performed as previously described [12]. The total IgA concentrations in feces were measured by using an enzyme-linked immunosorbent assay (ELISA) kit for quantitative analyses (Invitrogen, Waltham, MA, USA). Mucins were quantified by using a fluorometric assay for quantitative analysis of fecal mucin (Cosmo Bio Co., Ltd., Tokyo, Japan). Fecal total IgA and mucins were measured in accordance with the manufacturer’s instructions.

The OVA-specific IgA levels in feces were evaluated by ELISA, as described previously [18]. Fecal samples were diluted 10-fold with PBS and then centrifuged at 15,000× *g* for 15 min at 4 °C. The supernatants were subjected to specific IgA ELISA. Flat-bottomed microtiter plates were precoated with 100 μL of OVA (10 μg/mL) in a 0.1 M carbonate buffer (pH 9.6) and incubated overnight at 4 °C. After the wells had been washed with PBS containing 0.05% Tween 20 (PBS-T), a 1% bovine serum albumin (BSA)/PBS-T solution was added to each well, and the solution was incubated for 1 h at 37 °C. After washing each well three times with PBS-T, 100 μL of each fecal sample was applied to each well, and the mixture was incubated for 1 h at 37 °C. After washing each well five times with PBS-T, 100 μL of HRP-conjugated IgA (Southern Biotech, Birmingham, AL, USA) and anti-mouse IgE (diluted 1:5000 in 1% BSA/PBS-T) was added to each well, and the solution was incubated for 1 h at 37 °C. Each well was then washed five times with PBS-T, followed by the addition of 100 μL of *o*-phenylenediamine (0.4 mg/mL) in citrate-phosphate buffer (pH 5.0) containing 0.006% H_2_O_2_. The reaction was quenched with 50 μL of 2.5 M H_2_SO_4_ (aq.) after 7 min at room temperature, and color development was measured at 492 nm using a microplate reader (POWERSCAN^Ⓡ^HT; BioTek Instruments, Winooski, VT, USA).

### 2.6. Immunohistochemistry

Liver samples were fixed with 4% paraformaldehyde phosphate buffer solution for 7 days and then displaced with 30% sucrose/0.1 M phosphate buffer (pH 7.4) solution in 0.2% NaN_3_. After confirming that the tissue had settled in the solution, it was reduced using dimethylaminoborane and then freeze-embedded in an OCT compound (SECTION-LAB, Yokohama, Japan). The cut surface was covered with an adhesive film (Cryofilm type IIC9; SECTION-LAB), and frozen sections (10 μm) were prepared with a Leica CM3050S cryostat in accordance with the method of Kawamoto [19]. Immunohistochemical studies were performed using the commercially available ImmPRESS^Ⓡ^Horse Anti-Rabbit IgG Polymer Kit, Peroxidase, as described above and in accordance with the user guide of the manufacturer (Vector Laboratories, Newark, CA, USA). Endogenous peroxidase activity was blocked in 3% H_2_O_2_ for 10 min. After washing the slides twice with PBS for 5 min, non-specific binding was blocked in 2.5% normal horse serum from the ImmPRESS^Ⓡ^Horse Anti-Rabbit IgG PLUS Polymer Kit, Peroxidase Reagent Kit (MP-7801; Vector Laboratories, Inc.,, Newark, CA, USA) for 30 min. Excess serum was also removed from the samples. In the next step, the samples were incubated overnight with a primary rabbit polyclonal ovalbumin antibody (200-401-033, 1:300 dilution; Rockland, Pottstown, PA, USA) at 4 °C. Subsequently, the slides were washed twice with PBS for 5 min. They were then incubated for 2 h with horse anti-rabbit IgG polymer reagent. Next, the slides were washed for 2 × 5 min in PBS. After the buffer had been removed from the slides, they were incubated in 3,3’-diaminobenzidine solution for 5 min, followed by repeated washes in PBS. Stained tissue sections were viewed and photographed with NIS Elements software (Ver. 4.30) using an upright microscope (Nikon Eclipse Ti-U; Nikon Co., Tokyo, Japan).

### 2.7. Histology

Colon samples were fixed with 4% paraformaldehyde phosphate buffer solution for 7 days and then embedded in paraffin. Sections (4 μm) were cut and stained with periodic acid-Schiff (PAS)/Alcian blue (AB) for goblet cell enumeration. Stained tissue sections were viewed and photographed with NIS Elements software using an upright microscope.

### 2.8. Statistical Analysis

Data are expressed as the mean ± SEM. All statistical analyses were performed using IBM SPSS Statistics version 25.0 (IBM Japan, Tokyo, Japan). Within-group analysis was conducted by one-way analysis of variance (ANOVA). Tukey test or Games–Howell test was also performed as a post hoc test. Differences were considered to be significant, with *p*-values of less than 0.05.

## 3. Results

### 3.1. Body Weight, Feed Intake, and Dry Feces Weight

The results of body weight, feed intake, and dry feces weight on the day before the experimental period began and the last day of the experiment are shown in Table 2. There were no significant differences between groups in body weight or feed intake. However, the dry weight of feces was significantly higher in the Canna and OVA-Canna groups than in the Control and OVA groups (*p* < 0.01).

### 3.2. Measurement of Body Temperature

During systemic anaphylaxis, decreases in whole-body temperature are commonly observed [20,21]. In this study, rectal temperature was measured 30 min after the OVA challenge to evaluate anaphylactic symptoms (Figure 1). The rectal temperature in the OVA group was significantly lower than that in the Control and Canna groups (*p* < 0.01). Conversely, the OVA-Canna group exhibited a significant increase in this variable compared with the OVA group (*p* < 0.05), with no significant decrease compared with the Control and Canna groups. When body temperature was used as an indicator, the results suggested that canna starch ingestion suppressed anaphylactic symptoms.

### 3.3. Immunohistochemical Analysis Using Anti-Ovalbumin Antibody 

The absorption of OVA from the gastrointestinal tract by the liver was evaluated using immunostaining with anti-OVA antibodies in a double-blind manner. Figure 2 shows the leakage of OVA from the gastrointestinal tract into the liver. A darker brown color indicates that OVA was detected. In the OVA group, there was an increase in OVA accumulation in the liver compared with that in the Control and Canna groups. In contrast, the OVA-Canna group showed a decrease compared with the OVA group and exhibited findings comparable to those of the Control and Canna groups. These results suggested that canna starch ingestion prevents the absorption of OVA from the gastrointestinal tract by the liver.

### 3.4. Evaluation of Intestinal Barrier Function Using Fecal IgA and Mucin

OVA-specific IgA, total IgA, and mucin in feces were measured to assess intestinal barrier function (Figure 3). For all measures, higher values indicate higher intestinal barrier function. Daily fecal OVA-specific IgA was significantly elevated in the OVA and OVA-Canna groups compared with that in the Control and Canna groups (Figure 3A *p* < 0.05), but there was no difference between the OVA and OVA-Canna groups and no effect of canna starch intake. No differences in total IgA were observed between the groups (Figure 3B). Daily mucin was significantly elevated in the Canna and OVA-Canna groups compared with that in the Control and OVA groups (*p* < 0.01) and was increased by the consumption of canna starch (Figure 3C).

### 3.5. Effect of Canna Starch Intake on Intestinal Mucosal Barrier

Alcian PAS staining of colon tissue was performed to examine mucin in detail. Alcian PAS is used to differentiate neutral and acidic mucins within a tissue section. Acidic mucins are stained blue with AB, and PAS stains neutral mucins magenta. Tissues and cells that contain both neutral and acidic mucins stain varying shades of purple due to the binding of AB and the reaction with Schiff reagent. The colon tissue exhibited a consistent magenta and purple tint across all conditions. Notably, variations in purple areas, representing goblet cells, were observed among the four groups. Acidic mucus related to mucins and goblet cells were assessed in a double-blind manner (Figure 4). The results showed an increase in this staining in the OVA-Canna group compared with that in the Control, Canna, and OVA-Canna groups, indicating that the consumption of canna starch led to an enhanced capacity of colon goblet cells to produce mucins and increased intestinal barrier function due to the increase in mucins.

## 4. Discussion

The prevalence of food allergies is increasing, but no treatments are currently available, which has heightened interest in prebiotic strategies. We previously found that the intake of canna starch, which is rich in both amylose and resistant starch, increases barrier function in the intestinal tract and improves the intestinal environment [12]. These results suggest that canna starch could serve as a beneficial prebiotic. Previous reports indicated the prevention of food allergies in mice through the consumption of indigestible starch with prebiotic effects, including the simultaneous intake of human milk oligosaccharides [22] and short and long fructans [23]. Therefore, it is possible that the intake of canna starch does not allow allergens to enter the body or cause allergic symptoms. In this study, we thus focused on the effects of canna starch on food allergy model mice, paying particular attention to the intestinal tract.

In the type I allergy model mice, allergen administration increases vascular permeability, resulting in anaphylaxis with symptoms of decreased blood pressure and body temperature [20]. It is also reported that, in mice, OVA concentration in peripheral blood vessels peaks at 20–30 min [24]. Therefore, to determine the effect of canna starch on anaphylactic symptoms in food allergy model mice, rectal temperature was measured 30 min after administration of the allergen. The results showed that canna starch suppressed anaphylactic symptoms in mice. Various responses contribute to the development of type I allergies, and it is possible to inhibit allergic responses at different stages. These strategies may involve complete allergen degradation through enhanced digestion, preventing allergen absorption by prompting IgA production, suppressing allergic reactions by inhibiting IgE production and its binding to mast cells and also curtailing the production and release of mediators, including the signaling for mediator release. We previously showed that canna starch consumption enhances gastrointestinal barrier function, mainly in terms of mucin and IgA, in the gastrointestinal tract of mice [12]. Patients with food allergies have increased intestinal permeability due to defective intestinal barrier function, which correlates with the severity of clinical symptoms. This, in turn, suggests that intestinal permeability is related to the onset and severity of allergy [25,26]. We hypothesized that the suppression of anaphylactic symptoms in a mouse model of food allergy due to canna starch intake might be attributable to the inhibition of the uptake of allergenic OVA into the body. To assess this, we immunostained the liver with OVA antibodies to measure the amount of OVA that had accumulated in this organ after ingestion through the intestinal tract. The results indicated that canna starch intake inhibited the accumulation of OVA in the liver of food allergy model mice. Although it has been reported that increased allergen absorption exacerbates symptoms in rats [27], in the current study, a decrease in allergen uptake by canna starch ingestion in the food allergy model mice may be one reason for the suppression of anaphylactic symptoms.

While the primary role of the intestinal tract is the digestion and absorption of food, it is constantly exposed to a variety of microorganisms, including pathogens and foreign substances like chemicals, in addition to ingested food. To counteract these challenges, the intestinal mucosa is coated with a thick layer of mucus, which forms a non-specific barrier. The major component of this mucus is mucin, a high-molecular-weight glycoprotein primarily produced by goblet cells. Mucin is heavily glycosylated, and these glycoconjugates are believed to provide mucus with its viscosity and serve as a competitive binding site, preventing allergens and pathogenic microorganisms from attaching to the sugar chains on the surface of epithelial cells, thus inhibiting bacterial invasion and adhesion [28]. Against this background, we examined the effects of canna starch ingestion on intestinal barrier function in a mouse model of food allergy. We previously showed that canna starch intake in mice enhances mucin and IgA secretion in the intestinal tract [12]. In the present study using food allergy model mice, there were no changes in the production of total IgA with or without sensitization and canna starch ingestion. OVA-specific IgA was increased by sensitization but was not affected by canna starch intake. Meanwhile, mucin was increased in canna starch-fed mice with or without sensitization. In rats, an increase in mucin secretion due to the ingestion of dietary fiber has been reported, with Vahouny et al. [29] speculating that dietary fiber increases in bulk because it is not digested in the intestinal tract and that the physical stimulation of the intestinal tract leads to increased mucin secretion. The canna starch used in this study is rich in resistant starch, which, similar to dietary fiber, is not digested in the intestinal tract [12]. The observed increase in fecal weight after canna starch consumption may have augmented the bulk of intestinal contents, thus stimulating mucin secretion. In addition, *Akkermansia muciniphila* degrades mucin using acetic acid and propionic acid as energy sources while promoting epithelial cell development, which in turn promotes the differentiation of mucin-producing goblet cells and thus increases mucin levels [30,31]. We, therefore, evaluated the expression of goblet cells responsible for mucin production in the colon by immunostaining. Our results showed that canna starch intake increased the number of goblet cells and the mucin-producing capacity in the colon tissue of mice. Although it could not be clarified in this study, the increase in mucin may have been influenced by an elevated level of butyric acid, which could in turn lead to an increase in *A. muciniphila*. Future research will require comparison with other starch foods to determine whether these results are specific to canna starch, given that the study was limited to canna starch. In addition, there is a need for further analysis of the intestinal microbiota, organic acids, short-chain fatty acids, and the mucin gene muc2, the product of which is primarily secreted by goblet cells. Within the intestinal tract, one of the largest immune organs, various factors contribute to barrier function, including mucin, IgA, tight junctions, and immune cells. In humans, intestinal barrier function and intestinal permeability correlate with the severity of clinical symptoms, and intestinal permeability is thought to be related to the onset and severity of allergy. Intestinal permeability is maintained by various factors, one of which is mucin, and the results of this study suggested that mucin synthesis is related to intestinal health and allergy. In this study, the increase in mucin could potentially have enhanced the intestinal barrier function and inhibited the uptake of allergenic OVA.

## 5. Conclusions

This study examined the effect of canna starch ingestion on food allergy model mice. The results suggested the following: (1) oral administration of OVA suppressed anaphylactic symptoms; (2) liver uptake of OVA was decreased; and (3) the production of goblet cells and mucin in feces and colon tissue was increased. This suggested that canna starch may improve intestinal barrier function, inhibit the absorption of allergens, and suppress anaphylactic symptoms in this food allergy mouse model.

## Figures and Tables

**Figure 1 biomolecules-14-00215-f001:**
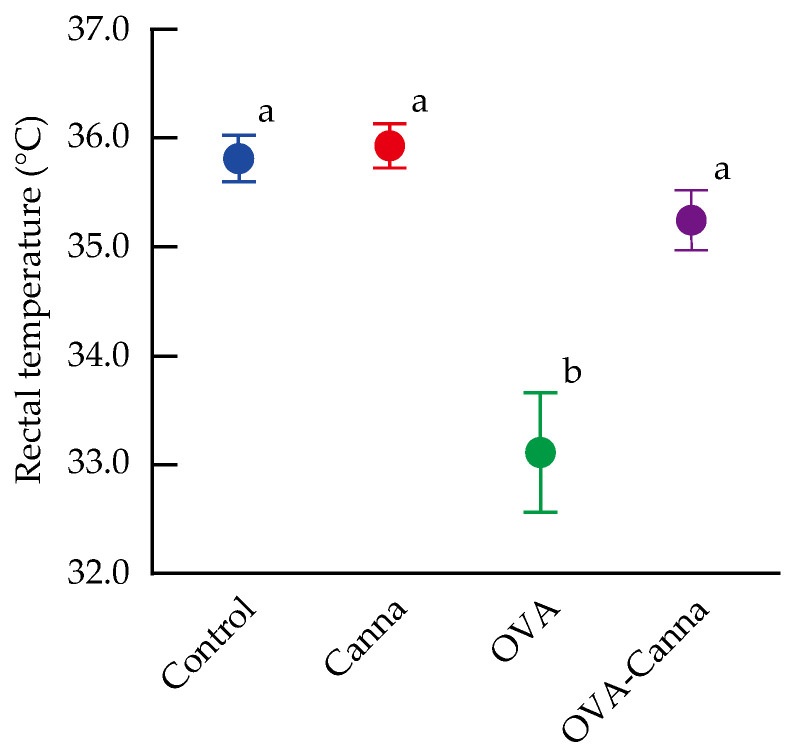
Allergic symptoms after oral challenge in mice sensitized in the presence of canna starch. On day 28, the rectal temperature of the mice was measured 30 min after OVA administration to evaluate anaphylactic symptoms. Data are expressed as the mean ± SEM (n = 8–10). Means without a common letter differ significantly (*p* < 0.05).

**Figure 2 biomolecules-14-00215-f002:**
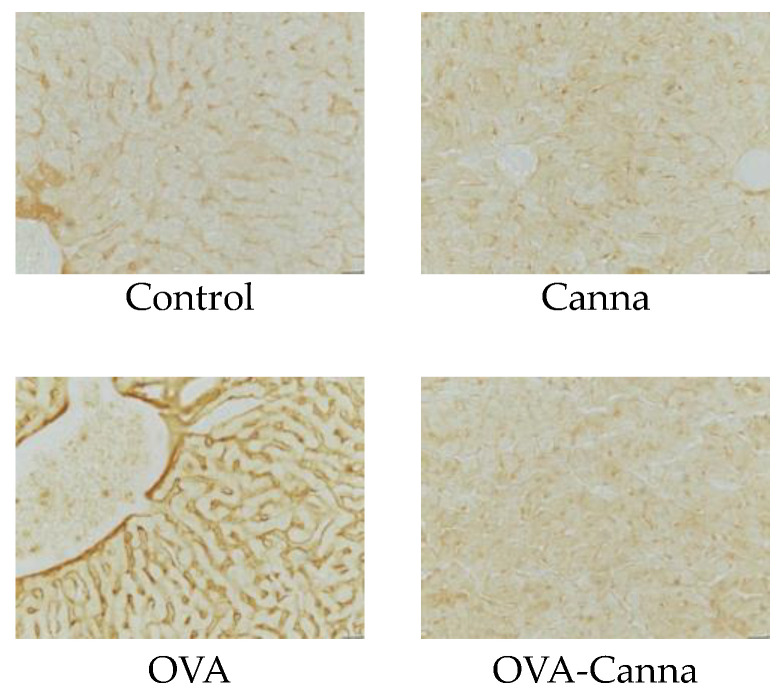
Immunohistochemical analysis of hepatic ovalbumin localization. After the evaluation of anaphylactic symptoms on day 28, the liver was collected, fixed, and used to prepare frozen sections. The images are representative of each group. Immunohistochemical staining of liver sections was used to determine the effects of canna starch on OVA localization (400×).

**Figure 3 biomolecules-14-00215-f003:**
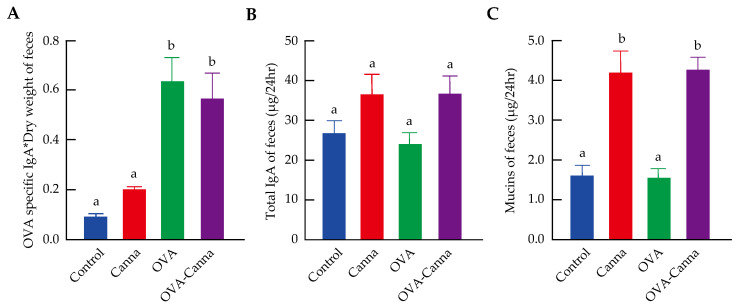
Effects of a diet containing canna starch on fecal IgA and mucins in mice: The feces collected between day 25 and day 28 were freeze-dried, and the following variables per dry weight were measured: (**A**) OVA-specific IgA levels in feces; (**B**) fecal total IgA amount in each group; and (**C**) number of fecal mucins in each group. Data are expressed as mean ± SEM (n = 8–10). Means without a common letter differ significantly (*p* < 0.05).

**Figure 4 biomolecules-14-00215-f004:**
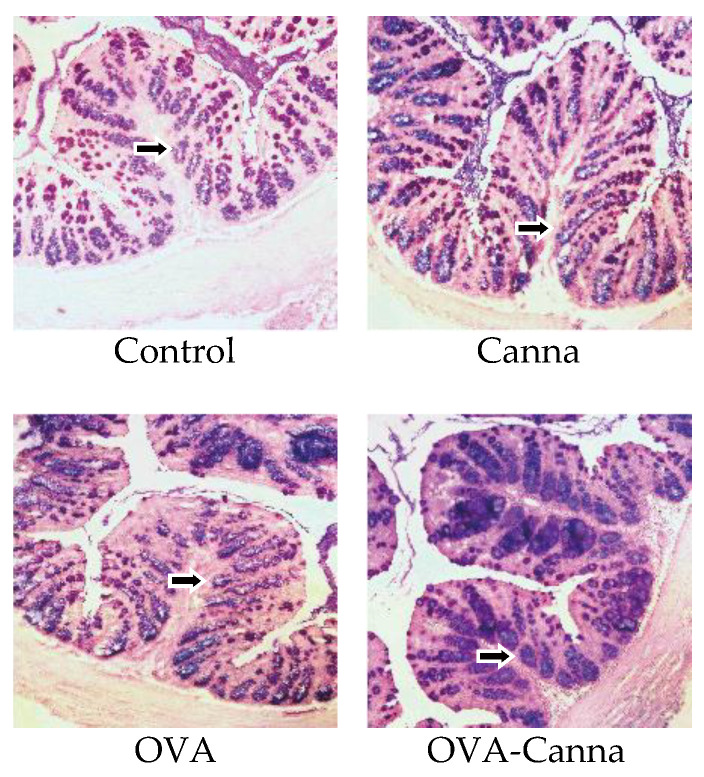
Colonic tissue subjected to Alcian PAS staining. After the evaluation of anaphylactic symptoms on day 28, the colon was collected, fixed, and used to prepare paraffin sections. Goblet cell production (arrows) could be observed at every segment. The images are representative of each group. Immunohistochemical staining of colon sections was used to determine the effects of canna starch on Alcian PAS staining (200×).

**Table 1 biomolecules-14-00215-t001:** Composition of the test diets (%).

Component	Control Diet	Canna Diet
Cornstarch	63.2	53.2
Casein	20.0	20.0
Corn oil (no additives)	7.0	7.0
Fiber	5.0	5.0
Mineral mix (AIN-93G-MX)	3.5	3.5
Vitamin mix (AIN-93VX)	1.0	1.0
L-Cystine	0.3	0.3
*tert*-Butylhydroquinone	0.0014	0.0014
Canna starch	0.0	10.0
Total	100	100

**Table 2 biomolecules-14-00215-t002:** Body weight gain, food intake, and dry weight of feces.

	Control	Canna	OVA	OVA-Canna
Body weight gain (g)				
Day 0	17.3 ± 0.3 ^a^	17.4 ± 0.3 ^a^	17.4 ± 0.3 ^a^	17.5 ± 0.3 ^a^
Day 28	18.3 ± 0.4 ^a^	18.9 ± 0.2 ^a^	19.5 ± 0.4 ^a^	19.3 ± 0.4 ^a^
Food intake (g/day)	2.74 ± 0.05 ^a^	2.74 ± 0.04 ^a^	2.68 ± 0.04 ^a^	2.69 ± 0.03 ^a^
Dry weight of feces (g/24 h)	0.20 ± 0.02 ^a^	0.44 ± 0.02 ^b^	0.20 ± 0.01 ^a^	0.46 ± 0.02 ^b^

Data are expressed as the mean ± SEM (n = 8–10). Means in the same row without a common letter differ significantly (*p* < 0.05).

## Data Availability

Data are contained within the article.

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
