# Peer review of "Canna Starch Improves Intestinal Barrier Function, Inhibits Allergen Uptake, and Suppresses Anaphylactic Symptoms in Ovalbumin-Induced Food Allergy in Mice"

_biomolecules, 2024, doi:10.3390/biom14020215_

Round 1
Reviewer 1 Report
Comments and Suggestions for Authors
This is an interesting topic about the improvement of intestinal barrier function of canna starch intake. However, the present study does contribute limited information due to the presence of some obvious issues with the way in which the authors have presented their material and the results. Specific comments are presented as follows
The article title does not match the content of main text because of the additional appearance liver immunohistochemical analysis.
Section 2.2 Animal and diets:
(1)Experimental design is not described clearly how the four groups of mice were fed with the control diets and the canna strach diets. In addition, do you mean OVA group and OVA-canna group of mice are fed with the OVA-control diet and OVA-canna diet respectively? If so, how to prepare the experimental diets inclusing OVA control and OVA-canna diets? How do you conduct the feeding trial? Were the four groups fed their respective diets, namely control diet, canna starch diet, OVA-control diet, and OVA-canna starch diet during the feeding period of 28 days? or were the four groups treated with or without OVA at the end of the feeding trial?
(2)Table1: The experimental diet contains 10% canna starch. How do you think about the suitability and effectiveness of the inclusion level of canna starch for the present study? There are inconsistent description about the control diets (AIN-93G only presents in table 1 and abstract) and canna diets (10% canna starch-added diet only presents in table 1 and bastract) in terms of the design of dietary treatments.
Section 2.4 Animal protocol:
(1)There is no description about the feeding conditions of mice and the determination of growth performance across the feeding duration of 28 days.
(2)I do not understand why authors perform oral OVA administration for all the tested mice including the control mice? Does this practice conflict with the experimental design (four experimental groups)?
(3)Please explain the reason for the inconsistent number of mice in different groups. The methods of immunostaining and Alcian Blue-PAS staining should be described in brief with citations.
Section 2.5. Analyses of fecal specific IgA, total IgA, and mucin:
(1)IgA may be rich in blood. Please explain why you determine Ig A of feces instead of the blood.
(2)line 136: The sentence "microliter plates were precoated with 100 ml of OVA in a 0.1 M carbonate buffer" is difficult to understand.
(3)line 140: The sentence "100 mL of each fecal sample was applied to each well" is hard to read.
Section 3.1. Body weight, feed intake, and dry feces weight:
(1)Table 2: You determined the weight of feces (dry basis) collected for 24 h, why? How to collect feces to determine daily fecal excretion.
Section 3.2. Inhibitory effect on anaphylactic symptoms using a mouse model
(1)There was limited specific information.
(2)The directionality of title of "3.2. Inhibitory effect on anaphylactic symptoms using a mouse model" is unclear.
Section 3.3. Effects of preventing absorption of OVA from the gastrointestinal tract by the liver:
(1)The title of "3.3. Effects of preventing absorption of OVA from the gastrointestinal tract by the liver" is not concise.
(2)There was limited specific information.
(3)There were no signs of Immunohistochemical differences that presented in Figure 2. Also the resolution of the images is low.
Section 3.4. Effect of canna starch intake on the intestinal environment:
(1)The title of "3.4. Effect of canna starch intake on the intestinal environment" does not match this content.
(2)There was limited specific information.
Section 3.5. Effect of canna starch intake on intestinal mucosal barrier
(1)There was limited specific information.
(2)Figure 4 was not found in the section "3.5 Effect of canna starch intake on intestinal mucosal barrier". According to colon tissue histology analysis, the method of PAS/Alcian blue is used for goblet cell enumeration. However, the captions of Figure 4 stress the colon tissue is treated with the immunohistochemical staining. Change differences of dietary treatments reflecting anaphylactic symptoms are not seen. What cells have undergone changes, and how are the changes related to intestinal mucosal barrier? A positive or negative effect of canna diet on allergical response was not described clearly. Figure 4 has low resolution and lacks arrows indicating specific cells.
Comments on the Quality of English LanguageEnglish needs improvement.
Reviewer 2 Report
Comments and Suggestions for Authors
The article provides valuable insights and presents the effect of Canna starch on improving intestinal barrier function and suspending anaphylactic symptoms in mice. The authors executed the study adequately; however, there are some concerns that require the author's attention.
1. Change the starting lines of the abstract (lines 19-20), and provide some introductory lines to start the introduction.
2. In the introduction, explain how the canna starch differs from the wheat, sweet potato, ginger, and corn potato starch. Also, emphasize this in the discussion to justify the selection of canna starch.
3. Is there any specific reason for the different number of animals in the groups (Line 94).
4. In Figure 2, it would be more informative if the author could quantify the results using Image J software or any other tool.
5. Author mentioned figure 8? (Line 227). But there is no Figure 8 in the MS; replacing this with Figure 3 is better.
6. Describe all the results more elaborately.
7. Add figure no. in section 3.5.
8. Rewrite the lines "However, there … …. burden (lines 43-44)", the meaning of the statement is not clear.
9. Rewrite the lines (line no. 99-100).
10. Statement in line no. 106 is not completed; therefore, rewrite the statement again. Similarly, the statement in lines 115-116 is not clear.
11. In all the figure legends, rewrite the statement "Means without ……significantly (p<0.05).
Comments on the Quality of English LanguageModerate english editing is required.
Reviewer 3 Report
Comments and Suggestions for Authors
This study provides encouraging but preliminary findings as it investigates the impact of canna starch on food allergies in mice. The mice were separated into groups, and some were given a diet that contained canna starch. Compared to the group that was fed only ovalbumin, the group that was fed both ovalbumin (to cause allergies) and canna starch experienced fewer severe allergic responses. Additionally, the canna starch groups showed better gut health. The study is restricted to a mouse model, despite the fact that these results imply that canna starch may improve intestinal barrier function and lessen allergy symptoms.
Given the promising results demonstrated in the study, indicating that canna starch may enhance intestinal barrier function and reduce anaphylactic symptoms in a mouse model of food allergy, publishing this article could contribute valuable insights to the field of allergy research and offer a foundation for further investigations into potential dietary interventions for allergy management, but before publication, I have a few questions for the authors:
How can the authors ensure that the observed decrease in anaphylactic symptoms and reduced ovalbumin (OVA) uptake in the liver in the OVA-Canna group is directly attributable to canna starch, and not to other variables or external factors?
The results of the study show that the OVA-Canna group's colon goblet cells produced more mucin. Is it possible to ascertain if this improvement is exclusively attributable to the use of canna starch and is not impacted by other dietary elements or environmental factors? Does this higher synthesis of mucin, especially in human individuals, connect with gut health in general and allergy resistance?
How can the authors link the observed rise in mucin production and decrease in allergen uptake to canna starch consumption alone, given the complexity of the intestinal immune system and its interactions with different dietary components, without considering the possible impact of other dietary factors or the larger gut microbiome? how do these findings translate to human physiology, given the differences between human and murine gut biology?
Round 2
Reviewer 1 Report
Comments and Suggestions for Authors
This paper is improved greatly, but there is some space to raise the quality of the paper.
(1) There are two sets of diagrams (Figure 4). Which one should be kept?
(2) The design in fact is a two-factorial trial design (canna and OVA treatments). In this sense, completing the analysis of effects of dietary canna on OVA may require following the procedure: comparing the results between OVA+canna group and OVA group, then analyzing the degree of improvement of canna on OVA. The control group is just a reference for canna group.
Reviewer 2 Report
Comments and Suggestions for Authors
The MS has been substantially modified. I recommend for the article publication.
Comments on the Quality of English LanguageNA
Author Response
Thank you very much for taking the time to review this manuscript.